# Comparison of Choi, RECIST and Somatostatin Receptor PET/CT Based Criteria for the Evaluation of Response and Response Prediction to PRRT

**DOI:** 10.3390/pharmaceutics14061278

**Published:** 2022-06-16

**Authors:** Kevin Zwirtz, Juliane Hardt, Güliz Acker, Alexander D. J. Baur, Marianne Pavel, Kai Huang, Winfried Brenner, Vikas Prasad

**Affiliations:** 1Department of Nuclear Medicine, Charité Universitätsmedizin Berlin, 13353 Berlin, Germany; kevin.zwirtz@mail.de (K.Z.); kai.huang@charite.de (K.H.); winfried.brenner@charite.de (W.B.); 2Institute of Biometry and Clinical Epidemiology, Charité Universitätsmedizin Berlin, 13353 Berlin, Germany; juliane.hardt@tiho-hannover.de; 3Department of Biometry, Epidemiology and Information Processing, WHO Collaborating Centre for Research and Training for Health in the Human-Animal-Environment Interface, University of Veterinary Medicine (Foundation) Hannover (TiHo), 30559 Hannover, Germany; 4Medical Information Management, Faculty of Information and Communication, University of Applied Sciences Hannover, 30459 Hannover, Germany; 5Department of Neurosurgery, Charité Universitätsmedizin Berlin, 13353 Berlin, Germany; gueliz.acker@charite.de; 6Department of Radiology, Charité Universitätsmedizin Berlin, 13353 Berlin, Germany; alexander.baur@charite.de; 7Department of Hepatology and Gastroenterology, Charité Universitätsmedizin Berlin, 13353 Berlin, Germany; marianne.pavel@uk-erlangen.de; 8Department of Endocrinology, University Hospital of Erlangen, 91054 Erlangen, Germany; 9German Cancer Consortium (DKTK), Partner Site Berlin, and German Cancer Research Center (DKFZ), 69120 Heidelberg, Germany; 10Department of Nuclear Medicine, University Ulm, 89081 Ulm, Germany; 11International Centers for Precision Oncology Academy, 65189 Wiesbaden, Germany

**Keywords:** peptide receptor radionuclide therapy (PRRT), Ga-68 DOTATOC, Ga-68 DOTATATE, PET/CT, RECIST, Choi, SUVmax

## Abstract

**Aim:** The most suitable method for assessment of response to peptide receptor radionuclide therapy (PRRT) of neuroendocrine tumors (NET) is still under debate. In this study we aimed to compare size (RECIST 1.1), density (Choi), Standardized Uptake Value (SUV) and a newly defined ZP combined parameter derived from Somatostatin Receptor (SSR) PET/CT for prediction of both response to PRRT and overall survival (OS). **Material and Methods:** Thirty-four NET patients with progressive disease (F:M 23:11; mean age 61.2 y; SD ± 12) treated with PRRT using either Lu-177 DOTATOC or Lu-177 DOTATATE and imaged with Ga-68 SSR PET/CT approximately 10–12 weeks prior to and after each treatment cycle were retrospectively analyzed. Median duration of follow-up after the first cycle was 63.9 months (range 6.2–86.2). A total of 77 lesions (2–8 per patient) were analyzed. Response assessment was performed according to RECIST 1.1, Choi and modified EORTC (MORE) criteria. In addition, a new parameter named ZP, the product of Hounsfield unit (HU) and SUVmean (Standard Uptake Value) of a tumor lesion, was tested. Further, SUV values (max and mean) of the tumor were normalized to SUV of normal liver parenchyma. Tumor response was defined as CR, PR, or SD. Gold standard for comparison of baseline parameters for prediction of response of individual target lesions to PRRT was change in size of lesions according to RECIST 1.1. For prediction of overall survival, the response after the first and second PRRT were tested. **Results:** Based on RECIST 1.1, Choi, MORE, and ZP, 85.3%, 64.7%, 61.8%, and 70.6% achieved a response whereas 14.7%, 35.3%, 38.2%, and 29.4% demonstrated PD (progressive disease), respectively. Baseline ZP and ZPnormalized were found to be the only parameters predictive of lesion progression after three PRRT cycles (AUC ZP 0.753; 95% CI 0.6–0.9, *p* 0.037; AUC ZPnormalized 0.766; 95% CI 0.6–0.9; *p* 0.029). Based on a cut-off-value of 1201, ZP achieved a sensitivity of 86% and a specificity of 67%, while ZPnormalized reached a sensitivity of 86% and a specificity of 76% at a cut-off-value of 198. Median OS in the total cohort was not reached. In univariate analysis amongst all parameters, only patients having progressive disease according to MORE after the second cycle of PRRT were found to have significantly shorter overall survival (median OS in objective responders not reached, in PD 29.2 months; *p* 0.015). Patients progressive after two cycles of PRRT according to ZP had shorter OS compared to those responding (median OS for responders not reached, for PD 47.2 months, *p* 0.066). **Conclusions:** In this explorative study, we showed that Choi, RECIST 1.1, and SUVmax-based response evaluation varied significantly from each other. Only patients showing progressive disease after two PRRT cycles according to MORE criteria had a worse prognosis while baseline ZP and ZPnormalized performed best in predicting lesion progression after three cycles of PRRT.

## 1. Introduction

Peptide receptor radionuclide therapy (PRRT) is an established treatment option for patients with progressive Grade 1, Grade 2 metastasized gastroenteropancreatic neuroendocrine tumors (GEP NET) [1,2,3]. This is largely attributed to (a) high linear energy transfer of short-ranged beta particles; (b) relatively long half-life of radionuclides resulting in continuous superfractionated radiation and (c) moderate to very high tumor radiation dose due to high density of somatostatin receptor expression (SSTR) on NETs [4]. Based on dosimetry models, in NET metastases and primary tumors radiation doses ranging from 10–340 Gy can be achieved [4,5]. However, previous prospective and retrospective studies have demonstrated that in most patients PRRT achieves stabilization of disease based on Response Evaluation Criteria in Solid Tumors (RECIST) rather than notable tumor size reduction (partial remission, PR), which is observed only in up to 18–30% of patients [1,2]. Interestingly, patients with PR were found to have similar overall survival (OS) rates as compared to patients with stable disease (SD) [6]. This indicates that response evaluation based on RECIST is not a reliable tool for slow-growing NET. Haug et al. and Gabriel et al. realized the drawbacks of RECIST and suggested PET (Positron Emission Tomography) based response evaluation criteria [7,8]. Both authors however reached contradictory conclusions. Whereas Haug et al. suggested SUVmax tumor/SUVmax spleen ratio (Standard Uptake Value) as a useful parameter for response assessment after PRRT, Gabriel et al. concluded that PET had no advantage over conventional anatomic imaging for assessing response to therapy at all [7,8]. Moreover, none of the previously published studies found a reliable baseline parameter for predicting disease progression or remission. 

Based on this knowledge, we realized that there is an urgent unmet need for suitable predictors and response assessment criteria for PRRT. This quest for finding appropriate disease-specific response assessment criteria is not unique to NET: mRECIST was developed specifically for hepatocellular carcinoma (HCC), and Choi for gastrointestinal stromal tumors (GIST) [9,10]. The Choi response parameter takes into account not only changes in tumor size but also in tumor attenuation. 

Attenuation, reflected by Hounsfield Unit on CT, is a measure of the density of the tissue. Some anticancer drugs like sunitinib, imatinib, etc. lead to a change in attenuation earlier as compared to a change in size [11]. Similarly, in usually well-vascularized NET, tumor cell density also plays an important role as the linear energy transfer (LET), or in other words, mass-energy-interaction, is directly proportional to the density of the targeted material or tissue. 

In this study, our main aim was to compare response evaluation with respect to overall survival (OS) in patients treated with at least two cycles of PRRT based on size (RECIST 1.1), attenuation (Choi), and somatostatin receptor expression (mod. EORTC) of tumor lesions. Our secondary aim was to search for other suitable predictors of lesion progression or remission based on baseline PET/CT characteristics. For this purpose, we assessed the performance of well-established parameters attenuation (HU), SUVmax, and SUVmean. Additionally, we evaluated a new combined parameter ZP, defined as attenuation (HU) derived from CT, multiplied with SUVmean of the respective lesion, derived from PET. 

## 2. Patients and Methods

### 2.1. Study Design and Ethical Clearance

GEP NET patients referred for PRRT from June 2011 to December 2015 at our centre were retrospectively analyzed. The decision to perform PRRT was taken in an interdisciplinary tumor board. Included in this analysis were patients (a) with progressive, histopathologically documented grade 1, grade 2 NET (according to the WHO classification) with at least one measurable lesion on CT or MRI, (b) treated with at least 2 cycles of PRRT with Lu-177 DOTATOC or Lu-177 DOTATATE, (c) imaged with Ga-68 DOTATOC or Ga-68 DOTATATE (SSTR) PET/3-phase contrast enhanced (ce) CT at baseline, (d) follow-up imaging with SSTR PET/CT performed within 3 months after the last cycle of PRRT, and (e) at least one 3-phase ceCT or SSTR PET/CT (3-phase) after 1st and 2nd PRRT cycles. The mean duration of follow-up after the first PRRT was 63.9 months (6.2–86.2 months). Figure 1 is showing the study design.

A total of 34 out of 69 GEP NET patients matched all the inclusion criteria. This retrospective analysis was approved by our institutional ethic committee (EA1/168/17).

### 2.2. Radiopeptide Treatment

The choice of peptide, DOTA-TATE or DOTA-TOC for PRRT was largely influenced by their availability. Lu-177 DOTA-TATE or Lu-177 DOTA-TOC was administered to the patients by slow intravenous infusion over 10–15 min. For renal protection, all patients received 1500 mL of an amino acid infusion (250 mL NaCl plus Lysine HCL 5% plus 250 mL L-Arginine HCL 10%) over 4 h. In 34 patients a total of 101 PRRT cycles were performed. Patients received a mean of 3 cycles (range 2–6) of PRRT at an interval of 10–14 weeks. Patients were imaged with SSTR PET/CT at baseline and at 10–12 weeks after PRRT cycle. Follow-up PET/CT was performed in some of the patients on the day of the PRRT. On an average, patients were given a mean activity of 6.58 GBq Lu-177 DOTATOC (*n* = 29) or Lu-177 DOTATATE (*n* = 72). 

### 2.3. Somatostatin Receptor PET/CT

In order to avoid any interference of long-acting somatostatin receptor analogs (SSA) on the SSTR expression of tumor cells, patients were imaged with SSTR PET/CT at least 4 weeks after the last injection of SSAs. Ga-68 was eluted from Ge-68/Ga-68 generators and labelled either with DOTATATE or DOTATOC according to the respective standard labelling procedure [12]. The selection of either DOTATATE or DOTATOC for imaging was purely based on the availability of the compound due to patent regulations. It has been demonstrated previously that there is no significant difference in the tumor uptake between these two tracers [13]. SSTR PET/CT was performed according to the European Association of Nuclear Medicine (EANM) Guidelines from 2010 [14]. Mean radioactivity injected was 100–120 MBq and the acquisition was performed 45–60 min after the injection of the radiotracer. All PET scans were acquired in a 3-dimensional acquisition mode on a Gemini TF 16 PET/CT system (Philips Medical Systems, Eindhoven, The Netherlands). The standard 3D-LOR (Line of Response) algorithm of the system software was used with default parameter settings to reconstruct transverse slices of 144 × 144 voxels with 4.0 × 4.0 × 4.0 mm^3^. For contrast-enhanced multi-phase CT performed at the time of PET/CT, 70–100 mL Ultravist 370 (Bayer Schering Pharma, Berlin, Germany) was injected intravenously and images were acquired using bolus tracking (threshold 100 HU) with a total delay of around 30 s after start of the injection for the arterial phase, 50 s for the porto-venous phase, and 70 s for late venous phase [15].

### 2.4. Image Analysis 

Target lesions were defined according to the specification of “Response Evaluation In Solid Tumors (RECIST 1.1)” [16]. Number of target lesions was also taken in accordance with RECIST 1.1. Additionally, the PET component was taken into consideration for differentiation between tumor vs. nontumor lesion. As far as possible, cystic lesions at baseline were not chosen as target lesions. As all patients had progressive disease prior to 1st PRRT, stable disease as well as partial remission and complete remission were defined as tumor response. 

#### 2.4.1. PET

The PET/CT images were analyzed by a junior scientist (KZ) under the supervision of an experienced board-certified radiologist (AB), and a nuclear medicine physician (VP). In case of discrepancy between these readers, a second nuclear medicine physician (WB) was involved in the final decision. Lesions seen on PET/CT were characterized as tumor tissue or metastases only if both the radiologist (AB) and a nuclear medicine physician (VP, WB), all with >15 years of experience in PET/CT, achieved a common consensus.

The somatostatin receptor expression in tumor and normal liver tissue was semi-quantitatively assessed by calculating the maximum and mean standardized uptake value (SUVmax; SUVmean). SUVmean for both the tumor region and the normal liver was determined by using a 40% isocontour on transverse attenuation-corrected PET slices. For lesion segmentation on PET, borders of lesion on CT were taken into consideration and in cases where the isocontour under- or overshot the border, PET lesion borders were manually adjusted. The SUVmean in the liver was taken as reference value. Thereafter the SUVmax of the tumor lesions was normalized using SUVmean of the liver according to the formula:normalized Uptake in tumor (SUVratio) = SUVmax Tumor/SUVmean liver

SUV was measured only for those lesions which were definitely positive by visual assessment, i.e., the uptake of the lesion was higher than the uptake of the normal surrounding tissue, and the lesion had a size of more than 10 mm in diameter.

PET-based response measurement was performed according to criteria defined by the European Organization for Research and Treatment of Cancer (EORTC) [17]. Adapted to the SSTR PET/CT-based imaging, this assessment was termed molecular response evaluation (MORE). The patient disease status was classified based on SUVmax: (a) progressive disease (PD) was characterized as >25% increase in SUVmax or any new lesions, (b) stable disease (SD) when change in SUVmax was between −25% and +25% according to the target lesions, (c) partial response (PR) was defined by minimum 25% decrease and (d) complete metabolic response (CR) when no lesions were visible on PET anymore. In addition, we also tested normalized SUVmax for response assessment by the aforementioned criteria.

#### 2.4.2. CT

Diameters of target lesions were measured in the longest cross-sectional dimension or in the short axis for lymph nodes at each time point of follow-up. The sum of all target lesion measurements was computed and the absolute and percent change between the pre-treatment and the follow-up scans were calculated. The attenuation of each lesion was measured in HU on venous phase CT after drawing a Region-of-Interest (ROI) in the lesion. The absolute and percentage changes in HU during the treatment were calculated.

It is important to note that there is a substantial difference in the definition of PR and PD on RECIST 1.1 and Choi. RECIST 1.1 classifies a lesion as partial remission if there is a decrease in diameter of ≥30% whereas according to Choi criteria lesions with a decrease in diameter of ≥10% and/or change in attenuation (HU) of ≥15% are considered as PR.

#### 2.4.3. ZP Parameter

In order to find a new suitable parameter for the evaluation of response to PRRT, we generated the new quantitative hybrid parameter ZP, which is based on the idea of the linear energy transfer (LET) principle. LET is directly proportional to the density of the material. Density is by definition mass per volume. Attenuation measured in CT using HU is dependent on density.

Radiation energy deposition in an area is dependent on the number of beta particles which is directly proportional to the somatostatin receptor density, and the density of the material. Therefore we decided on the following formula:ZP (Target) = SUVmean (Target) × HU (Target)
ZPnormalized (Target) = normalized SUVmean (Target) × HU (Target)
where HU (attenuation) is an indirect measure of density of the tumor in CT and SUVmean is a surrogate for the average receptor expression density in a tumor that was hypothesized to represent the mass of radiolabeled peptide in the tumor. 

ZP was calculated only in patients receiving contrast-enhanced CT during SSTR PET/CT in order to avoid potential miscalculations associated with misregistration of PET data with diagnostic CT data performed separately on different scanners. 

For comparable response evaluation, we decided to use a classification analog to MORE for the new parameter ZP: progressive disease (PD) was defined as an increase of ≥25% in the product of HU and SUVmean, whereas a reduction of ≥25% determined a partial response (PR). Complete response (CR) was defined as no detectable lesions on CT and PET. Any lesion that did not match these criteria was described as stable disease. Table 1 is showing the comparison of different response assessment parameters.

### 2.5. Statistical Analyses 

SPSS v13 (IBM) was used for statistical calculations. Absolute and relative frequencies were calculated for categorical variables. According to histograms and normality plots, a non-parametric distribution of image-derived parameters was assumed and descriptive parameters are given as median and range (minimum–maximum). Differences between two unpaired groups were analyzed using the Mann–Whitney U test. Receiver operating characteristic (ROC) analysis was performed to determine the association of a metric and a binary variable. The point on the ROC curve with the minimal distance to the point with 100% sensitivity and 100% specificity was defined as the optimum cut-off value. The data for pseudoprogression is described in frequency and percentage, values are given in median and range.

The significance of the difference between OS measured on Kaplan Meyer (KM) Curve for response assessment parameters (RECIST, Choi, MORE, ZP) was tested using log-rank test. In addition, we performed Pearsons’ Chi-squared test to determine the independence amongst different response parameters for patients progressing under PRRT in comparison to those not progressing. For all univariate analyses as well as two-sided tests, a *p*-value of less than 0.05 was considered significant.

## 3. Results

### 3.1. Demography, Histopathology, Previous Therapy

Out of 69 patients screened for this study, 34 patients (F:M 23:11; mean age 61.2 y; SD ± 11.98) fulfilled the inclusion criteria; 35 patients were excluded because of the following reasons: two patients had only bone lesions (*n* = 2), no follow-up PET/CTs was available in 18, no target lesions in 11 patients, and in 4 patients CT was performed without contrast media. 

In the majority, the primary tumor was located in the ileum (*n* = 11) and pancreas (*n* = 9). According to WHO 12 patients were classified Grade 1 and 22 as Grade 2 NET. Eleven patients showed values of Ki-67 up to 2%, and 23 patients showed values between 3–20%. 

Prior to PRRT, patients received either one or multiple treatments (mean: *n* = 2; range: 1–3): 22 patients underwent surgery, 20 were treated with somatostatin analogs, 18 received chemotherapy, 2 were treated with external beam radiation therapy and 2 patients received transarterial chemoembolization of liver lesions- these lesions were not selected as target lesions. Table 2 is showing patients’ characteristics.

### 3.2. PET/CT

PET/CT was performed at baseline and at a mean of 3.2 months after every PRRT cycle. In these 34 patients, 77 lesions were evaluated by follow-up PET/CT scans (2–8 per patient): 50 liver lesions, 21 lymph node metastases, 2 lung, and 2 ileum lesions, 1 lesion in the thymus, and 1 rectal lesion. 

### 3.3. Response (RECIST 1.1, Choi, MORE, ZP)

Based on RECIST 1.1, 85.3% of patients achieved a response (SD [*n* = 9], PR [*n* = 20] and CR [*n* = 0]) after the last therapy cycle. Based on Choi, 64.7% had a response (SD [*n* = 10], PR [*n* = 12], CR [*n* = 0]. MORE criteria suggested response in 61.8% (SD [*n* = 13], PR [*n* = 8], CR [*n* = 0]), whereas ZP based evaluation showed response in 70.6% of patients (SD [*n* = 14], PR [*n* = 10], CR [*n* = 0]). There was a weak but significant correlation between Choi and RECIST 1.1 as tested by Phi coefficient, Cramer’s V and Spearman’s ρ (value 0.562; *p* 0.001), because both response evaluation methods are primarily based on size. There was significant correlation between ZP and Choi criteria (value 0.231; *p* 0.043). MORE, i.e., SUV-based based response evaluation did not correlate with Choi or RECIST 1.1 (value 0.082; *p* 0.496 and value 0.156; *p* 0.364). There was also no correlation of ZP with any other evaluation method RECIST (value 0.057; *p* 0.574) and MORE (value 0.191; *p* 0.094) See Table 3. 

### 3.4. Prediction of Response to PRRT

Progression according to RECIST 1.1 for individual lesions was taken as the gold standard for response evaluation. The predictive value of baseline parameters i.e., diameter, SUV max, SUV mean, HU as well as ZP and ZPnormalized for prediction of response after three therapy cycles of PRRT was assessed. ZP and ZPnormalized were the only parameters found to be predictive of lesion progression according to RECIST 1.1. With a cut-off value of 1201, ZP achieved a sensitivity of 86% and a specificity of 67% (AUC 0.75; *p* = 0.037), while ZPnormalized reached a sensitivity of 86% and specificity of 76% at a cut-off value of 198 (AUC 0.77; *p* = 0.029). Figure 2 is showing the ROC for the prediction of response to three therapy cycles based on baseline imaging parameters. Examples are demonstrated in Figure 3 and Figure 4.

### 3.5. Pseudoprogression

Pseudoprogression was defined as an initial increase in diameter of a lesion by more than 10% but a decrease in size after subsequent PRRT cycles.

Nine out of 77 lesions in 7/34 patients showed an initial mean increase in the size of 15% (11 to 39%) after the first PRRT cycle. These lesions subsequently showed a mean decrease in size of 8% (+8 to −26%) indicating a pseudoprogression (see Figure 5).

### 3.6. Overall Survival 

Median OS in the total cohort was not reached. In a univariate analysis of tumor grade (according to WHO), RECIST, Choi, MORE and ZP patients with progressive disease after the first therapy cycle did not show a worse prognosis compared to responders. 

Similar results were observed for the RECIST and Choi assessments after the second PRRT cycle. Patients having progressive disease according to MORE after the second cycle of PRRT however were found to have a significantly shorter OS (median OS for responders not reached, for PD 29.2 months; *p* 0.015). Patients with progressive disease after the second PRRT according to ZP appeared to live shorter as compared to those achieving response, however, the difference was not significant (median OS for response not reached, 47.2 months for PD, *p* 0.066).

Figure 6 is showing the Kaplan Meier curve (Figure 6A) for overall survival of the whole cohort, (Figure 6B,C) show the overall survival of patients progressive after the second therapy cycle vs. those showing response based on MORE and ZP criteria respectively, (Figure 6D) patients having mean ZPnormalized > 198 showed better prognosis as compared to patients with ZPnormalized < 198. 

## 4. Discussion

In our study, we focused on baseline imaging for the prediction of response to PRRT and prognosis of overall survival. To the best of our knowledge, this is the first study where RECIST 1.1 has been compared with both Choi and SUVmax-based response criteria (MORE) in patients treated with PRRT. Previous research has shown that the radiation dose delivered to tumor tissue is maximum in the first therapy cycle, while radiation-induced changes in the tumor and tumor microenvironment result in decreased doses to the tumor in subsequent treatment cycles [18,19,20]. On the other hand, the possibility of pseudoprogression should also be kept in mind especially during the first cycle of treatment. A previous study of the Rotterdam group reported pseudoprogression in 10% of patients [21]. In our study, we observed that 11% of the lesions which initially showed an increase in size after the first cycle subsequently decreased in size during further PRRT cycles (Figure 5). This could be one of the explanations why we observed a higher prognostic power of the interim staging after the second PRRT cycle in comparison to restaging after the first cycle. Patients progressing after two therapy cycles according to the MORE criteria revealed a shorter overall survival in comparison to patients who achieved a response. Interestingly, tumor grade in the univariate analysis did not turn out as a prognostic factor for PRRT outcome. More importantly, changes in two-dimensional size and tissue density (HU) which form the basis of RECIST 1.1 and Chois criteria, respectively, were neither predictive of disease progression after the first nor the second cycle of PRRT.

Regarding the ongoing discussion on appropriate predictors of response to PRRT, our results, albeit in a small cohort of patients, open up the possibility to use SUVs in future studies. Equally important seems to be the exploring of new and compound parameters based on SSTR PET/CT. We previously demonstrated that somatostatin receptor heterogeneity is both predictive and prognostic [22]. In this study, we explored the new response evaluation ZP parameter, the product of tumor SUVmean and tumor tissue density derived from CT. Although the parameter was not prognostic after the first PRRT cycle, it did suggest some value in patients after the second PRRT cycle: patients with PD according to the ZP tend to live shorter than patients achieving response although the difference was not significant probably related due to a limited number of patients. It remains to be seen if this parameter is robust in further prospective clinical trials. 

Delivery of an adequate radiation dose to lesions is dependent on interactions of radiation with matter, and thus, the density of tumor cells as well as on the number of available somatostatin receptors for actively binding the radiolabelled peptide. Previous studies have shown that the higher the somatostatin receptor density the better the response is to PRRT [6]. Haug et al. demonstrated that baseline SUVmax was a predictor of response to PRRT [8]. In the absence of a reliable benchmark to define responders vs. non-responders, we instead choose CT-based progression in tumor size according to RECIST 1.1 for defining our cut-off because size is used as a biomarker in most cancer treatment studies and considered as the gold standard. In our analyses, the best predictors for the progression of a given lesion during PRRT were ZP and ZPnormalized. Previous studies have shown that PRRT induces significant measurable changes in tumor size in up to 30% of patients. Interestingly, patients with partial response reveal a similar progression-free survival as patients with stable disease or minor response [6]. Therefore, Gabriel et al. tried to elucidate the usefulness of PET-based criteria for response evaluation in PRRT [6]. In their study, they did not find any correlation between changes in pre-post therapy SUVmax (ΔSUVmax) and TTP (time to progression) and, therefore, concluded that ΔSUVmax is not a useful parameter for PRRT response assessment. These findings of Gabriel et al. were partly supported by Haug et al. who reported that ΔSUVmax did not predict time to progression [8]. However, Haug et al. reported that patients with decreasing SUVmax tend to have longer TTP. This probably explains why we observed higher overall survival in patients responding to PRRT according to MORE after two therapy cycles as MORE also relies on changes in SUVmax. 

Treatment of GEP NET is ever-evolving due to the approval of several new drugs and positive phase III clinical trials in recent years [23,24]. Whereas surgery still remains the only curative treatment option, metastasized GEP NET patients are treated with systemic therapy alone or in combination with locoregional treatment and/or surgery [23,24]. The choice of second systemic treatments in a patient progressing under first-line treatment is often challenging due to partially overlapping indications of the second-line treatment options [23,24]. The transition from first- to second-line treatment in most clinical settings is governed by documentation of morphological progress based on CT or MRI. The CT or MRI-based response assessment in solid indolent tumors however is often hampered by the fact that it needs several months or longer before true positive changes in two-dimensional tumor size become quantifiable [25]. This necessitates sequential imaging at 3–6 month intervals, leading to increased radiation burden and economic burdens. This issue is further complicated by the fact that second- and third-line treatments have their own additional side effects. On the other hand, high-grade side effects of PRRT like severe nephrotoxicity and/or hematotoxicity occur in only 2–3% of patients. This necessitates the appropriate selection of patients and sequencing of therapies which ultimately influences overall survival. In this study, PET/CT-based response evaluations based on MORE and ZP criteria were found to have prognostic value. For slowly growing tumors like NET, PFS is a better endpoint as compared to overall survival for deciding on the efficacy of an anticancer drug. However, the retrospective setting, relatively low number of patients, heterogeneous cohort, different line of treatments prior to PRRT, different tumor biology, as well as inconsistent time-interval between successive imaging for follow-up after last PRRT cycle made the data almost impossible to compare and interpret the additional value of ZP, ZPnormalized, HU, size and SUvmax/mean in predicting PFS. 

## 5. Limitations

The retrospective nature and limited number of patients in this study are the main limiting factors. ZP has been assessed on the basis of a small number of patients, which could potentially bias the results. In current clinical practice, in the majority of the centers response to PRRT is assessed after completion of all therapy cycles. That is why the extent of the impact of the results discussed in this manuscript on prognoses after PRRT can only be ascertained when other centers also perform interim staging after two to three therapy cycles. In addition, as iodinated contrast also contributes to the density of the lesion measured on CT, future studies should be performed with native CT to assess the real impact of contrast enhancement on the HU and ZP factor.

## 6. Conclusions

In this explorative study, we showed that Choi, RECIST 1.1, and SUVmax-based response evaluation varied significantly from each other. Amongst all the criteria, MORE performed best: patients with progressive disease after two cycles of PRRT according to MORE criteria had a significantly worse prognosis. Baseline ZP and ZPnormalized performed best of all investigated parameters in predicting lesion progression after three PRRT cycles. Future prospective studies on GEP NET tumors with sufficient statistical power are needed to further validate both PET-derived criteria MORE and ZP.

## Figures and Tables

**Figure 1 pharmaceutics-14-01278-f001:**
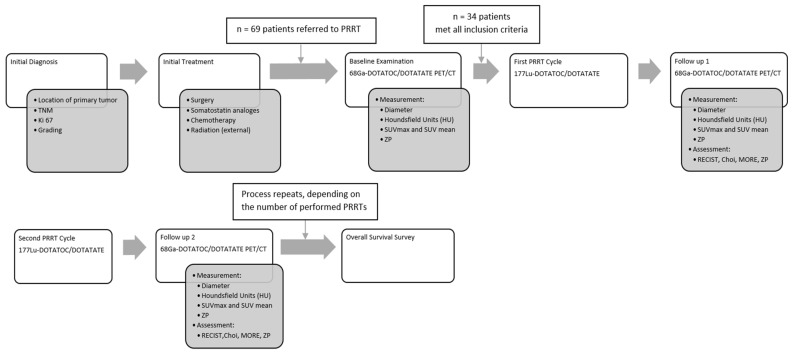
Study design.

**Figure 2 pharmaceutics-14-01278-f002:**
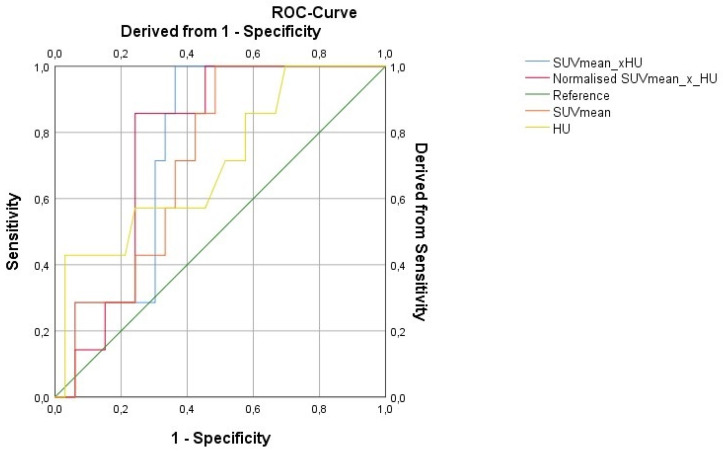
ROC for response prediction 3 therapy cycles based on baseline imaging parameters.

**Figure 3 pharmaceutics-14-01278-f003:**
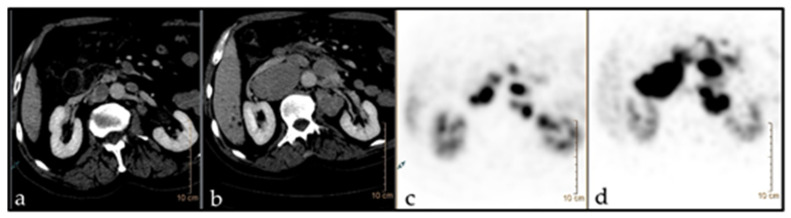
^68^Ga-DOTATATE PET/CT imaging in a 66-year-old patient (m) with NET of unknown origin. TNM n.a. (G2). The patient underwent chemotherapy with somatostatin analogues. Target: LN aortocaval. Baseline ZP: 2117. (**a**,**c**) before PRRT (diam.: 4.3 cm; SUV: 38.5); (**b**,**d**) after PRRT (diam.: 1.9 cm; SUV: 18).

**Figure 4 pharmaceutics-14-01278-f004:**
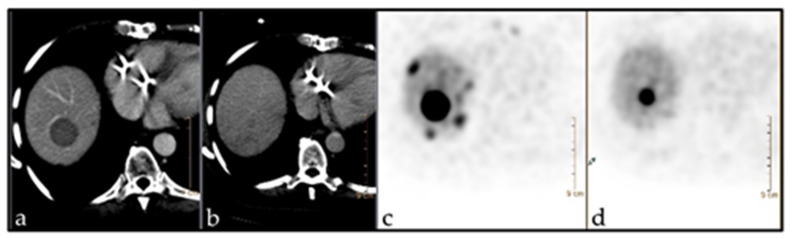
^68^Ga-DOTATOC PET/CT imaging in a 64-year-old patient (f) with NET of the terminal Ileum pT4, pN1, pM1 (G2). The patient underwent surgical intervention and chemotherapy with somatostatin analogues. Target: Metastasis in liver seg. VIII. Baseline ZP: 3524. (**a**,**c**) before PRRT (diam.: 2.8 cm; SUV: 61.2); (**b**,**d**) after PRRT (diam.: 1.2 cm; SUV: 28.1).

**Figure 5 pharmaceutics-14-01278-f005:**
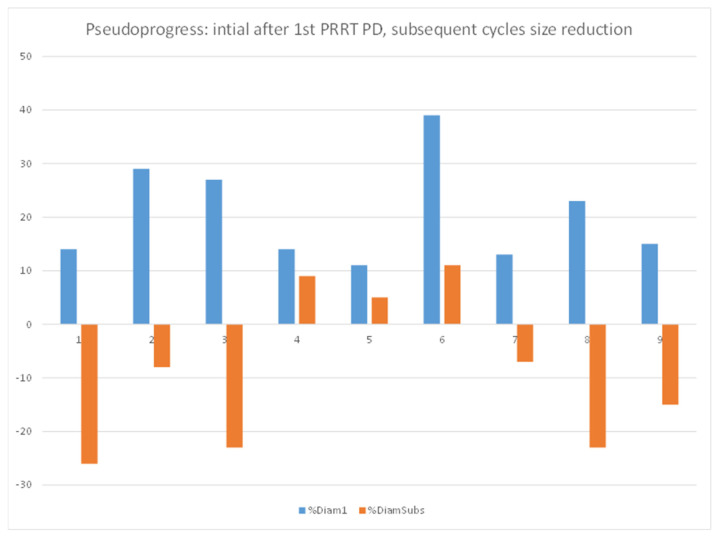
Pseudoprogression.

**Figure 6 pharmaceutics-14-01278-f006:**
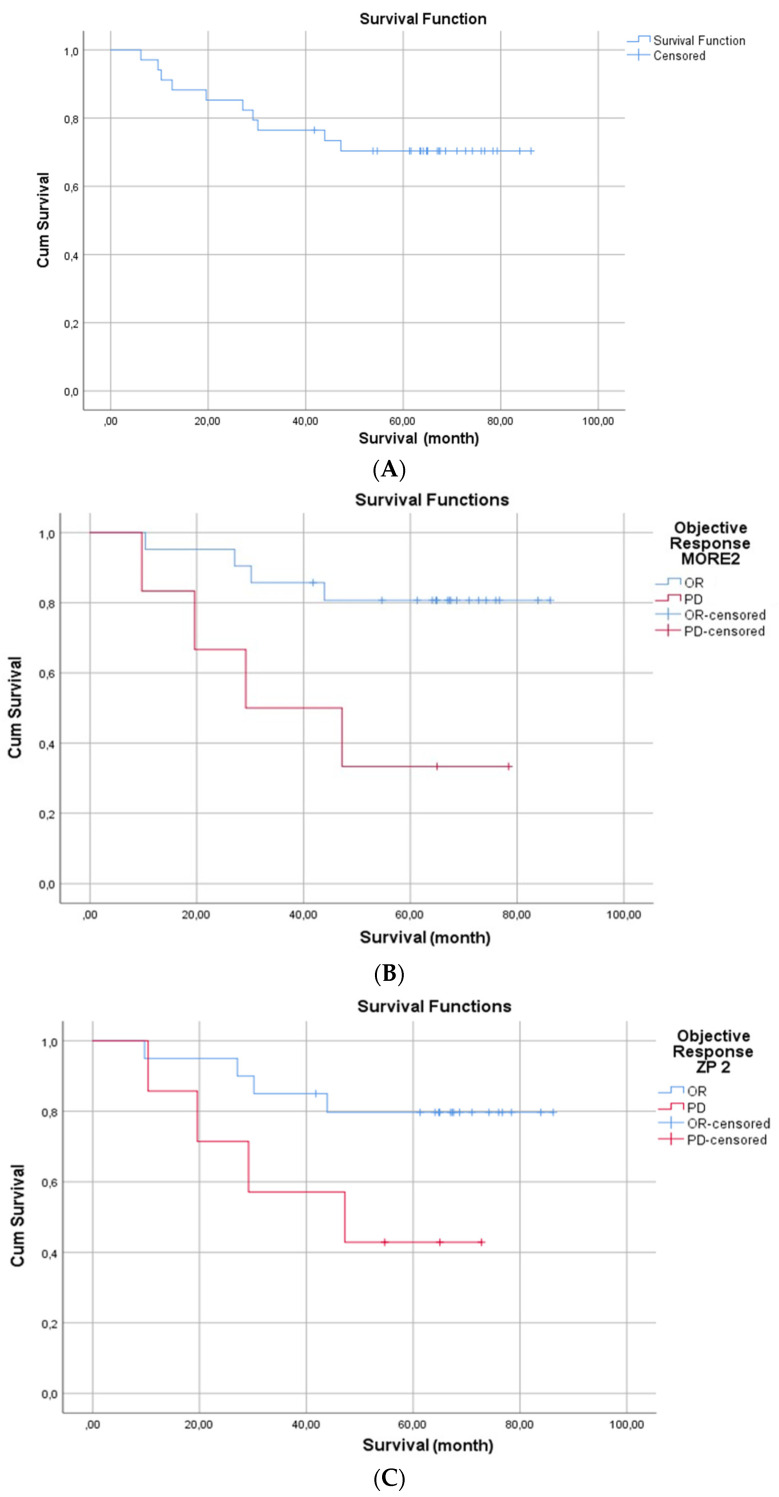
(**A**) Kaplan Meier curve for overall survival (OS) of the whole cohort; (**B**,**C**) OS of patients progressive after the second therapy cycle vs. those showing response based on MORE and ZP criteria respectively; (**D**) OS based on ZPnormalized cut-off value 198.

**Table 1 pharmaceutics-14-01278-t001:** Comparison of different response assessment methods.

		WHO	RECIST	Choi	MORE	ZP
**Non PD** (non-progressive disease)	**CR** (complete response)	Complete disappearance of all disease manifestations at an interval of at least 4 weeks	Disappearance of all lesions and no new lesions	Disappearance of all lesions and no new lesions	Complete disappearance of uptake in all lesions	No lesions detectable in CT or PET
	**PR** (partial remission)	Greater than or equal to 50% decrease in tumor size	≥30% reduction in the sum of the greatest diameter and no new lesions	≥10% decrease in the greatest diameter or a ≥15% decrease in tumor density (HU) and no new lesions	≥25% reduction in the sum of SUVmax after more than one cycle of treatment	≥25% reduction in the product of SUVmean and HU
	**SD** (stable disease)	Increase or decrease in tumor size of less than 25%	Does not meet the criteria for complete response (CR), partial response (PR), and progressive disease (PD)	Does not meet the criteria for complete response (CR), partial response (PR), and progressive disease (PD)	Does not meet the criteria for complete response (CR), partial response (PR), and progressive disease (PD)	Does not meet the criteria for complete response (CR), partial response (PR), and progressive disease (PD)
**PD** (progressive disease)	**PD** (progressive disease)	Greater than 25% increase in tumor lesions and/or appearance of new foci of tumor	≥20% increase in the sum of the greatest diameters or at least one new lesion	≥10% increase in the greatest diameter and does not meet the criteria for partial response (PR) Or at least one new lesion	≥25% increase in the sum of SUVmax or at least one new lesions	≥25% increase in the product of SUVmean and HU

**Table 2 pharmaceutics-14-01278-t002:** Patients and study characteristics.

Patients Characteristics (*n* = 34)	
Age in years, mean (SD)	61.19 (11.98)
Sex, male: female ratio	23:11
Median time interval between first diagnosis and first PRRT in months, mean (range)	32.35 (1.3–165.4)
Target lesions per patient, mean (range)	2.2 (1–4)
**PRRT**	
Mean number of therapy cycles (range)	3 (2–6)
Radiopeptide used (177 Lu-DOTATOC/DOTATATE)	29/72
**PET/CT Acquisition Data**	
Total number of PET/CT analyzed	130 (2–8 per patient)
Number of lesions evaluated	77
Time interval between PRRT and PET/CT in months, median (range)	3.2 (2.6–6.2)
Injected activity in GBq, mean (SD)	6.58 (0.87)
Time interval between tracer injection and acquisition in min, mean (SD)	60.3 (6.2)
**Follow-up**	
Median duration of follow-up from first PRRT in months, mean (range)	63.88 (6.2–86.2)
Median duration of follow-up after last PRRT in months, mean (range)	50.33 (1.9–69.0)

**Table 3 pharmaceutics-14-01278-t003:** Response distribution.

	nonPD		PD
**RECIST 1.1**	**29**		**5**
	CR	0	
	PR	20	
	SD	9	
**Choi**	**22**		**12**
	CR	0	
	PR	12	
	SD	10	
**MORE**	**21**		**13**
	CR	0	
	PR	8	
	SD	13	
**ZP**	**24**		**10**
	CR	0	
	PR	10	
	SD	14	

## Data Availability

The data presented in this study are available on request from the main author.

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
