# Peer review of "Comparison of Choi, RECIST and Somatostatin Receptor PET/CT Based Criteria for the Evaluation of Response and Response Prediction to PRRT"

_pharmaceutics, 2022, doi:10.3390/pharmaceutics14061278_

Round 1

Reviewer 1 Report

I would like to thank the authors for a novel idea to combine CT and PET parameters in order to predict response.

Although I do have some conserns:

- unlike the title the paper many addressess two issues, respons prediction and repsons assessment, could you make the title to reflect this?

- the evaluation of repsons prediction is only limitted to sensitivity and specificity, how where the cut-off chosen for these parameters. Can you chose an univariate analysis of these and other clinically relevant parameters concerning respons prediction? and possibly multvariate (some will not be indepenant ofcourse).

- the hyperdensity on ceCT of neuro-endocrine tumours is at least partialy based on the vascularisation of the tumour and the decrease that is depicted in the Choi criteria is also less vascularization. If you would like to make a link with density related energy transfer in therapy, it would seem more logical to use non-contrast enhanced CT.

Then again, a simillar mechanism as in GIST, still might apply as far as repons assessment is concerned, so I would suggest to leave the link to LET and possible therapy effect out.

- please provide some information on symptomatic tumours, markers like chromogranin A etc if available

- fig 1. Primarius is German, please translate as primary tumor / tumour (either US or UK English)

- please reformat table 3. All cathegories to the left, no borders around indivudual cells inside the table

- is possible higher resolution figures (3, 4, 5 and 6). They will not be readable after print. 

- remove overlap of the graphsin fig 6. The title of the last graph is only partially visible. 

- The x-axis in fig 6 should read Overall Survival or progression free survival???

- The legend in fig 6 is unclear ORR (objective respons rate???)

Reviewer 2 Report

The authors present a very interesting retrospective analysis of radiographic response inpatients with well differentiated neuroendocrine neoplasm  treated with lutetium. I agree with the authors that at the current time response assessment is limited to RECIST which may not be the most optimal measurement for measuring response to radioactive therapy.

The abstract clearly explained the background, methods, highlights of the results and their conclusion.

The results and methods were clearly reviewed/explained.

Due to the significant differences in measurement approaches, the statistical analysis was appropriate.

I agree with their assessment that the new assessment ZP tool may have some merit and deserves further evaluation in a larger cohort.

The one issue I had was that this analysis was based on early image analyses which is not a practice undertaken in routine care. Many don’t restage with imaging until after the second treatment, and some not until after completion. This clinical practice limits the utility of this analysis and new tool. It would be interesting to see if their results change if restaging imaging results taken after 2 and 4 treatments has the same prognostic utility.

I think this work is interesting and should be published. Minor grammatical errors need to be corrected, and authors should recognize that the frequency of image assessment is not in line with standard clinical practice which will limit it’s utility.
